# Vitamin D Serum Levels in the UK Population, including a Mathematical Approach to Evaluate the Impact of Vitamin D Fortified Ready-to-Eat Breakfast Cereals: Application of the NDNS Database

**DOI:** 10.3390/nu12061868

**Published:** 2020-06-23

**Authors:** Wim Calame, Laura Street, Toine Hulshof

**Affiliations:** 1StatistiCal BV, Strandwal 148, 2241 MN Wassenaar, The Netherlands; 2Kellogg’s Europe, Orange Tower, Media City UK, Salford, Manchester M50 2HF, UK; Laura.Street@marks-and-spencer.com (L.S.); toine.hulshof@gmail.com (T.H.)

**Keywords:** vitamin D, vitamin D intake and status, vitamin D fortification, breakfast cereals, National Diet and Nutrition Survey

## Abstract

Vitamin D status is relatively poor in the general population, potentially leading to various conditions. The present study evaluates the relationship between vitamin D status and intake in the UK population and the impact of vitamin D fortified ready-to-eat cereals (RTEC) on this status via data from the National Diet and Nutrition Survey (NDNS: 2008–2012). Four cohorts were addressed: ages 4–10 (*n* = 803), ages 11–18 (*n* = 884), ages 19–64 (*n* = 1655) and ages 65 and higher (*n* = 428). The impact of fortification by 4.2 μg vitamin D per 100 g of RTEC on vitamin D intake and status was mathematically modelled. Average vitamin D daily intake was age-dependent, ranging from ~2.6 (age range 4–18 years) to ~5.0 μg (older than 64 years). Average 25(OH)D concentration ranged from 43 to 51 nmol/L, the highest in children. The relationship between vitamin D intake and status followed an asymptotic curve with a predicted plateau concentration ranging from 52 in children to 83 nmol/L in elderly. The fortification model showed that serum concentrations increased with ~1.0 in children to ~6.5 nmol/L in the elderly. This study revealed that vitamin D intake in the UK population is low with 25(OH)D concentrations being suboptimal for general health. Fortification of breakfast cereals can contribute to improve overall vitamin D status.

## 1. Introduction

Vitamin D is pivotal in the well-being of every individual. Low concentrations of this vitamin in the body are considered to impact the incidence of various health conditions such as poor bone health [1,2,3]. With respect to the situation in the UK, the Scientific Advisory Committee on Nutrition (SACN) [4] has provided a major overview of the current vitamin D status as well as defining recommendations for adequate 25-hydroxy-vitamin D (25(OH)D) concentrations in the general population. The committee stated that both Ultraviolet B (UVB) exposure from sunlight as well dietary intake are the main suppliers of vitamin D. Up to 35% of total vitamin D intake is from meat and meat products (including fish), while milk and milk products too are key contributors, especially in children [4]. Overall (fortified) cereals and cereal products contribute 13 to 20% of the UK population’s vitamin D intake [4,5]. 

Typically, the nutritional status of vitamin D is determined via the serum concentration of circulating 25(OH)D [6]. These serum concentrations are found to be below those recommended for the entire population, in children [7], adults and the elderly [8,9]. Even applying a low serum threshold of 25 nmol/L, SACN [4] clearly states that 30–40% of the UK population are below this concentration in winter versus 2–13% in summer, underlining the importance of season in determining vitamin D status. 

Various approaches are being pursued in order to enhance the population’s poor vitamin D status, such as supplementation [10] and fortification [11]. Recommended intake levels of vitamin D have been defined by SACN UK, with a Reference Nutrient Intake (RNI) of 10 μg per day [4] to ensure adequate concentrations of nutrient status. SACN UK reports threshold concentrations of 25 nmol/L for 25(OH)D below which poor musculoskeletal health becomes a serious issue [4]. Again, the main conclusion to be drawn is that vitamin D intake is too low to safeguard sufficient concentrations of 25(OH)D in the body.

It is considered beneficial to enhance vitamin D intake in the daily diet, especially in seasons or circumstances that result in low sunlight exposure to the skin. Moreover, it is believed that high intake of vitamin D will yield relatively high serum concentrations [10,11,12]. Other factors, such as age, have also been reported to affect serum 25-hydroxyl-vitamin D concentrations [13]. The question remains, to what extent do higher vitamin D intake levels provide a better health status? Vitamin D was monitored as part of the National Diet and Nutrition Survey (NDNS), which provides information on health-related issues throughout the UK population and in which both vitamin D intake and status are measured. The outcome from the 2008/2009–2011/2012 data clearly revealed that the level of intake of vitamin D in all age groups within the UK population was far too low to meet the current recommendations for EAR (estimate of the average requirement) or RNI (Recommended Nutrient Intake). The NDNS 2008–2012 data were used in the current manuscript to gain further insight into the consumption data of vitamin D and to identify opportunities of how to optimise vitamin D status concentrations on a national level.

We selected ready-to-eat cereals (RTEC) as a vitamin D fortification food vehicle in a mathematical model in order to estimate the impact of food fortification on vitamin D status. We choose RTEC because they are a staple food in the UK, with an average penetration (proportion of people buying in the past year) up to 97% over the past 10 years and also frequently bought over the course of a year with a range between 22.5 and 24 buying occasions per person per year over the past 10 years (information kindly provided by Kellogg’s). Moreover, SACN [4] reports that 13–20% of the total dietary intake of vitamin D in the general population comes from cereal and cereal products. This demonstrates that RTEC are familiar and consumed by almost all individuals in the population, with a high frequency as well [14]. Another important issue for selecting RTEC is the stability of vitamin D in the food vehicle. Vitamin D fortification is done via vitamin D2 (ergocalciferol) and/or vitamin D3 (cholecalciferol). The stability of at least ergocalciferol is largely affected by the water activity: the higher the activity the lower the stability [15]. Since the vehicle (RTEC) will be in a dry form stability is expected to be of no concern. Obviously once dissolved in a liquid matrix (milk/water) it is advised to consume the product as soon as possible to avoid a potential loss in functionality. Overall, it has been observed that children and teenagers consuming breakfast cereals had significantly higher daily intakes of the fortified nutrients as well as better status for some of these nutrients [16] than those not consuming breakfast cereals. At the time of the data collection for the NDNS 2008–2012 a limited number of RTEC was fortified with vitamin D in the UK, at present this is still the case for the majority. In the fortification modelling we have used the distribution in vitamin D2 (ergocalciferol) and vitamin D3 (cholecalciferol) fortification and supplementation as present in the database. Unfortunately, no information on the actual type of vitamin D taken could be obtained, except which products were concerned: biscuits, cereal products (including chocolate-enriched products) and dairy products (ranging from low-fat butter to yoghurt). 

The main aim of the present study was to provide an overview on the intake and status of vitamin D in the UK population in four age cohorts of subjects surveyed in NDNS (2008–2012). Subsequently, the study tries to estimate the potential contribution of vitamin D fortification of RTEC to the change in vitamin D status in these age cohorts. The survey data allowed us to construct a mathematical model correcting for various factors, such as UV exposure and age to estimate the potential increase in status due to additional intake of vitamin D via RTEC. The four age cohorts used in the present study are: children (4–10 years, *n* = 803), adolescents (11–18 years, *n* = 884), adults (19–64 years, *n* = 1655) and elderly (from 65 years onwards, *n* = 428). In the UK population both intake and status of vitamin D is too low to support the health of the consumer satisfactorily. The relationship between intake and status showed an asymptotic relationship, while consumption of vitamin D fortified breakfast cereals yielded an improved status, especially in the elderly. 

## 2. Materials and Methods 

The various data files as collected by the NDNS were obtained from the UK Data Archives (NatCen, University of Essex, Colchester, Essex, UK). Information was selected from the NDNS Year 1–4 files (combined rolling program 2008/09–2011/12), which is an ongoing survey in the UK reviewing the diet and health-related issues of a representative sample of the UK population. Data are presented in cohorts defined by certain criteria e.g., age and gender [17]. The methods of the NDNS have been reported previously in full [17] and are briefly summarised here. Participants were recruited by a stratified random sampling technique using household postcodes from the Postcode Address File (a list of all the addresses in the UK). From each household, up to one adult and one child were invited to take part and written consent was obtained from adult participants and the parent/legal guardian of child participants. Participants completed a 4-day food diary to record all food and drink consumption and a face-to-face interview to obtain information on sociodemographic characteristics, lifestyle, and physical activity. Measurements of height and weight were also taken. Where consent was provided, participants were visited about 8 weeks later by a nurse and provided a blood sample to assess biochemical indices of nutritional status. Government bodies in Wales, Scotland and Northern Ireland are funding additional recruitment in their countries to facilitate region specific information. 

Within the current manuscript four cohorts were fully included: age ranges 4 to 11 years (children: *n* = 803), 11 to 19 years (teenagers: *n* = 884), 19 to 65 years (adults: *n* = 1655), and 65 plus (elderly: *n* = 428). Because some NDNS data were missing due to the lower participation for blood sampling than for nutrient intake, the total number of observations per parameter per cohort is not constant. Therefore, the number of measurements for the vitamin D status markers is substantially lower than for the equivalent nutrient intake measures. In order to obtain sufficient power when comparing and analyzing the nutrient status values, data of males and females have been combined in various analyses and indicated as such.

Ethical approval for the NDNS study (Ref. No. 07/H0604/113) was obtained from the Oxfordshire Research Ethics Committee. Ethical approval for our secondary analyses on the anonymized data was not required.

### 2.1. Definitions, Levels of Intake and Rationale for Level of Fortification

Within the current manuscript RTEC are defined as the summation of two cereal sub-categories from the NDNS database: ‘high fibre breakfast cereals’ and ‘other breakfast cereals’. The definition of high fibre breakfast cereals is: all breakfast cereals with non-starch polysaccharide (plant cell wall components of dietary fibre: Englyst fibre) of 4 g/100 g or more (including muesli, shredded wheat and porridge) while ‘other breakfast cereals’ include breakfast cereals with non-starch polysaccharide (Englyst fibre) of less than 4 g/100 g (including: Corn Flakes, Coco Pops as well as breakfast cereal and Nutri-Grain bars).

With respect to the required levels of intake, RNI has been taken into account in the current manuscript. The values were obtained from the latest SACN report [4].

The level of fortification of 4.2 μg of vitamin D per 100 g of breakfast cereals, as used in the current manuscript, is based on the assumption that a breakfast should provide approximately 25% of daily energy and nutrient requirements of vitamin D, depending on age and gender [18]. The official minimum labelling RDA for vitamin D in the UK is currently 5 μg per day [19]. Twenty-five percent of 5 μg equals 1.25 μg. A portion of 30 g breakfast cereals (on a dry weight basis) should provide this amount which results in fortification with 4.2 μg per 100 g. This number has been used in the various fortification models.

### 2.2. Measurement of Vitamin D Intake and Status in the NDNS Database

To calculate the vitamin D (D2 + D3) consumed for each food vehicle, the D2 and/or D3 content was multiplied by the quantity of the food item consumed and adjusted for the level of fortification. Unfortunately no detailed information on quantities of vitamin D2 and vitamin D3 of the item consumed was provided in the final data set. Nutrient status determined by 25-hydroxy-vitamin D concentrations was obtained from a blood sample after an overnight fast drawn by a nurse visiting the participant at home. The obtained sample was analysed via LC-MS/MS methodology, as described previously [20]. Blood sampling was conducted ~8 weeks after the collection of food intake data. 

### 2.3. Data Analysis

Throughout the study, survey analysis mode has been applied in the various statistical analyses, a mode in which the primary sampling unit was identified via the parameter “area” and the various strata via that of “gor” (Governmental Office Region). The clusters are an important modality in the above-mentioned mode. In the NDNS study these were considered as being clusters without replacement, no finite population correction had been conducted, because of the relatively low numbers of participants as compared to that in the total population. No weight factor was applied for various reasons, most importantly due to applying non-survey analyses after which the outcome of the various analyses could be compared. By using information on sampling units, strata and clusters the various analyses were more balanced with respect to representation by participants than by using that on weight factor, shown particularly in the relationship between daily vitamin D intake and vitamin D status. However, it is noted that weight factor has been calculated and was based upon information retrieved via an online source from the Office for National Statistics. The information was obtained from the 2009 data per strata used in the current research (http://www.statistics.gov.uk/statbase/). In some cases, the impact by weight factor was checked and was found to have minimal impact on the outcome, including the standard error. Estimation of population outcome variables, as well as regression analysis were all performed in survey analysis mode, this provided details of means and standard error. In the regression analysis confounding factors, such as age, have been taken into account too. In principal, the dependent parameter, concentration of 25(OH)D, is associated with various independent parameters; age, gender, BMI, time of the year, daily intake of vitamin D, daily intake of RTEC, daily intake of other dietary factors, but also economic factors (economic status). The analysis was performed in a stepwise approach, checking the model’s fit via F test. Repeated analysis was not performed due to the design of the survey. Associations between various parameters were analysed for significance via t-distribution and F test (the latter as an evaluation to which extent the variation observed is covered by the model used). The adjusted Wald test was applied to indicate a potentially significant difference between groups of people within certain parameters [21]. 

The relationship between time of year (UV exposure) and vitamin D status is described by the second order polynomial functions (function *y* = a*x*^2^ + b*x* + c, with *x* being time in months). As a result of this, the maximum vitamin D status per cohort can be calculated using the first derivative of the observed function. The peak concentration of 25(OH)D obtained in any month is expressed as numerical data with the various months as integers: 1: January, 2: February, etc. Variation coverage was enhanced by expressing modeling time (in months) as *x* and *x*^2^, as compared to cosine/sine function [22].

The calculation of serum values as a result of fortification modeling of RTEC, as defined previously, has been performed by applying the obtained formula describing the association between vitamin D in daily diet and concentrations of 25(OH)D. They were calculated while keeping the UV exposure information (month of sampling) constant. As a control the same people were used without fortification via the same formula. The difference between with and without fortification has been analysed by applying survey mode to ensure that the sampling design did not affect the outcome.

Outlier analysis was performed using Grubbs test [23], and once identified these outliers were removed from the respective analysis. Throughout the study applying two-sided evaluation and a *p*-value of 0.05 was considered to demonstrate significance.

All analyses were done in STATA, version 12 (Statacorp, College Station, TX, USA) and in GraphPad Prism, version 6 (LaJolla, CA, USA). Data handling was performed in Excel (Microsoft, Redmond, WA, USA) and checked within STATA, version 12. 

## 3. Results

### 3.1. Descriptive Information of Data from the NDNS Database before the Application of the Model

Table 1 lists the descriptive values of intake of vitamin D in μg per day per cohort and per gender. It can be observed that the absolute amount of vitamin D intake is higher with increasing age within the various age cohorts: the intake is highest in the elderly people.

As non-supplement consumers get older, the mean intake of vitamin D increases from 2.03 to 3.85 μg per day in males and from 1.89 to 2.76 μg per day in females, the males being consistently higher than the females. Among supplement consumers the percentage of takers is the lowest in teenagers (about 5–6%) followed by children and adults (14–19%) and the highest in elderly (24–32%). Among supplement consumers there is no increase in vitamin D intake over lifetime: the males mean range is between 5.92 to 9.43 μg per day and the females between 6.04 and 9.73 μg per day, with no difference between genders. 

Except for the boys 4–10 years old, from the mean values it is observed that in all supplement consumers the vitamin D intake from the diet (without supplementation) was higher compared to that in the non-supplement consumers. This suggests that those who take supplements already have a better vitamin D intake from their daily diet than the non-supplement takers. 

With respect to the intake of vitamin D based on 2008 to 2012 NDNS data, throughout the various cohorts the percentage of people below the recommended level of 10 µg per day (UK SACN) was extremely high, ranging between 98% to 100%. Supplementation of vitamin D was found to lower this percentage slightly, between 86% and 99%, nevertheless intake levels remained extremely low, irrespective of age and gender.

Table 2 shows combined (for power reasons) males and females 25-hydroxyvitamin D serum concentrations per age cohort. Approximately 20% of the children provided blood samples, this sampling percentage increases to ~35% in teenagers then to ~47% in those persons above 18 years of age. Data indicate that there is no substantial difference in the vitamin D status between the various cohorts, the highest value is noted for the age cohort 4–10 years and the lowest for the 65 plus cohort, however, there is no clear age-dependent effect. Considering the percentage of people below 25 nmol/L (defined as deficient in vitamin D status) or below 50 nmol/L (defined as insufficient in vitamin D status) these percentages are the lowest in children, while these numbers are constantly higher in adolescents, adults and the elderly. In the latter three cohorts, more than 20 % are defined as deficient and more than 60 % defined as insufficient. In children, the prevalence is 10% and 48% respectively.

### 3.2. Association between Daily Vitamin D Intake and Serum Concentrations of 25(OH)D (Data from the NDNS Database before Application of the Model)

Table 3 presents the relationship between the actual amount of vitamin D consumed per day, including supplementation, and serum concentrations of 25(OH)D. It is important to acknowledge the relatively low amount of observations in serum concentrations overall. To enhance the power, the outcome of both genders per age cohort were combined. The modeling calculations (Figure 1) show that a clear mean plateau concentration was reached in all cohorts, ranging between approximately 52–56 nmol/L, except for the elderly (who demonstrated a mean plateau concentration of ~83 nmol/L). Children are responding rapidly to higher vitamin D intakes in contrast to that observed in the elderly. The 2008–2012 NDNS results suggest that a higher intake of vitamin D to increase serum concentrations of vitamin D is needed in the elderly than in children. However, in the elderly the increase in 25(OH)D reaches a higher plateau concentration. Overall, a clear positive relationship exists between levels of vitamin D intake and concentration of 25(OH)D in serum. 

### 3.3. Impact of Time in the Year (Amount of UV Exposure) on Vitamin D Status (Data from the NDNS Database before Application of the Model)

The impact of UV exposure on vitamin D status is clearly observed in all cohorts. Please be aware that to establish this relationship, data of all people were used, including the non-RTEC consumers (to enhance the power). For this relationship, RTEC consumers are not the limiting factor since this parameter was not used in the statistical analysis. For every cohort, a parabolic (second order) regression curve can be obtained with a peak around month 7 to 8 (July–August). Since the curve follows a polynomial (second order) function (*y* = a*x*^2^ + b*x* + c, with x being time (in months) a clear optimum can be established (see Table 4). Peak concentrations are expressed as numerical data with 1: January, 2: February, etc. All models showed a significant (*p* < 0.001) fit following the second order polynomial function, variation coverage was significantly better as compared with cosine/sine functions. All cohorts demonstrated peak serum concentrations in the months of July to September, indicating an association between vitamin D status and amount of sun exposure. There is no difference in these peak concentrations between males and females or between age groups. Overall, taken all males and females together of all age cohorts used in the 2008–2012 NDNS study, the peak concentration of serum vitamin D is established in early August.

### 3.4. Potential Fortification Impact Using 4.2 μg of Vitamin D per 100 g of RTEC on Recommended Intake Levels of Vitamin D per Day (Mathematical Modeling Results)

Applying the obtained models (as described in Table 3) allows the contribution of fortification with 4.2 µg of vitamin D per 100 g of RTEC towards reaching SACN values to be calculated (Figure 2). For power reasons and due to the absence of a gender impact, the 4 cohorts are shown with both genders combined. 

The regression line shows the mean level of intake; 50% of the observations are below and 50% are above the actual regression curve. The outcome demonstrates that to reach 50% of the recommended level of daily vitamin D intake (10 μg), a portion size of ~60 g of RTEC is needed in the age groups 4–10 and 11–18, while ~34 g for adults between 19 and 64 and ~5 g for the elderly (65 and older). Moreover, since a substantial part of RTEC in the UK is recently fortified with 2.5 μg per serving (instead of 1.25 μg), the portion sizes to reach 50% are 30, 17 and 2.5 g respectively.

### 3.5. Potential Impact of Fortified Breakfast Cereals Consumption on Vitamin D Status (Mathematical Modeling Results)

Table 5 lists the calculated serum concentrations of 25-hydroxyvitamin D of RTEC with and without fortification with 4.2 μg per 100 g, with the restriction that UV exposure is kept constant. For obvious reasons only the RTEC consumers are included. In all cohorts, for both genders fortified RTEC intake results in an increase in serum concentrations, being the lowest in children and the highest in the elderly. For teenagers and the elderly, serum vitamin D increase is higher in males than in females with *p* < 0.002 and *p* < 0.05, respectively. This difference is related to the amount of RTEC consumed, being higher in males than in females in all cohorts, and becoming significantly different in teenagers (*p* < 0.0001) and in the elderly (*p* < 0.004). As observed for the 25(OH)D serum concentrations, the amount of RTEC consumed increases with age, being the lowest in children (~30 g per day) after which it increases to approximately 40 g per day in the adults and 65 g per day in the elderly. 

## 4. Discussion

The results of the 2008–2012 NDNS study clearly demonstrate that intakes of vitamin D in the UK are far too low throughout all age ranges taking into account SACN reference nutrient intake (RNI) values. In association with this, the serum concentration of vitamin D (25(OH)D) is relatively low, depending on age, gender and UV exposure. The most important factor determining these serum concentrations is UV exposure of the skin while daily vitamin D intake contributes significantly to 25(OH)D concentrations. RTEC represent an important part of the daily diet in the UK population and the intake of vitamin D can be increased by daily consumption of fortified RTEC. The relationship between daily intake and concentrations of 25(OH)D was found to follow an asymptotic (pseudo-Hill) equation establishing clear mean plateau concentrations, increasing from around 50 nmol/L in children towards ~80 nmol/L in the elderly. In the latter, the plateau concentration was reached at higher vitamin D intake levels than in children. It was calculated that RTEC fortified with 4.2 μg per 100 g vitamin D contributes significantly to the daily intake of vitamin D and thereby would enhance serum concentrations as well. Both increased healthy sun exposure and higher fortification of RTEC are viable strategies to improve vitamin D status. 

The outcome of this study is in good agreement with the intake recommendations SACN set for the UK population [4] which note that that dietary intake next to UV exposure is the best strategy to optimise vitamin D status. Fortification with 4.2 μg per 100 g of RTEC on a daily basis was calculated to increase 25(OH)D concentrations from ~1.0 in children to ~6.5 nmol/L in the elderly. 

Acknowledging that 13 to 20% of dietary intake of vitamin D is represented by fortified cereals and cereal products [4,5] these products are already an important candidate to increase vitamin D status. Both meat/meat products, milk/milk products and fat spreads are substantial sources of vitamin D. Dietary intake becomes of pivotal importance to maintaining healthy vitamin D status, especially when there is limited opportunity for adequate skin exposure to sunlight. Since we do not have detailed information on the relative distribution between ergocalciferol (vitamin D2) and cholecalciferol (vitamin D3) the relative distribution in fortification as present in the data of NDNS was applied. Importantly it might be worthwhile to consider fortification with cholecalciferol more than with ergocalciferol since upon consumption the former yields higher serum levels of 25(OH)D than the latter [24], but then this should be clearly labelled. Ergocalciferol is derived from plant (mushrooms) origin while cholecalciferol from animal origin, potentially leading to vegan issues. 

The optimal 25(OH)D concentration is still to be determined and is dependent on the physiological benefit ranging from bone health to insulin management [25,26,27]. Wolpowitz [26] evaluated which concentrations of 25(OH)D are necessary for specific health benefits: ranging from above 50 nmol/L for skeletal health to more than 100 nmol/L to reduce high blood pressure. In 2016, SACN [4] evaluated the threshold serum 25(OH)D concentration and found below 25 nmol/L to be indicative of increased risk of poor musculoskeletal health. The 2008–2012 survey intake and serum concentration data suggest that, in all cohorts, plateau concentrations of 25(OH)D are being reached, increasing with age. However, in children, even at 95% CI concentration of this plateau values of ~60 nmol/L are being observed, while at older age, values of 90 nmol/L have been obtained, this has not been regularly observed in the 2008–2012 NDNS data. Therefore, the age of the individual is of importance to determine the adequate or optimal vitamin D concentration in serum. 

High concentrations of serum 25(OH)D are as problematic as low concentrations [28]. A Swedish cohort study in older men found a 50% higher mortality rate associated with concentrations below 64 nmol/L and above 98 nmol/L [29]. This U-shaped response curve has been noted by other studies [30]. To demonstrate the impact by the fortification as used in the present calculations the percentage of persons above either 75 nmol/L or 100 nmol/L is calculated applying the mathematical model (Table 6). For the analysis we have only used the RTEC consumers, since it might be argued to which extent non-consumers will start eating the fortified product at all. Again, it should be noted that the amount of data on which the mathematical model is based is relatively low. The outcome shows that only in the elderly a substantial increase is observed in persons with 25(OH)D levels above 75 nmol/L. Realizing that this might be associated with an increase in comorbidity the benefit should be weighed against the disadvantage.

Excessively high concentrations can be toxic, adversely affecting health in individuals, and should be avoided [31,32]. The tolerable upper intake levels of vitamin D for children aged 4 to 10 years is 50 μg per day and for males and females over 10 years old is 100 μg per day. 

A clear association between age and concentration of vitamin D in serum is observed in the present study. This is of importance since certain conditions are also linked with age, such as the incidence of certain types of cancer [25]. A reduction in prostate cancer and colon cancer is suggested at vitamin D serum concentrations of above 72 and 80 nmol/L respectively, concentrations currently being reached in the elderly. On the other hand, bone health is linked with concentrations above 50 nmol/L and within the current study this is in line with the mean plateau concentrations in all age groups. Increasing the concentration to a sufficient 50 nmol/L is feasible, leading to improved bone health. In the current study the mean plateau 25(OH)D concentrations are ~50 nmol/L and therefore it would be beneficial to increase these for optimising “overall” health in the population which can be achieved simply by increasing daily vitamin D intake. Obviously, unsafe high concentrations should be avoided [4], however applying the 4.2 μg per 100 g RTEC as calculated in the current study has been shown not to lead to excess concentrations, as shown in Table 5 and Figure 2 of the current manuscript. 

Unfortunately, the amount of observations in the nutrient status was relatively low and to improve the power of the analysis information of both genders were combined. However, throughout the various age ranges the status was always lower in females compared to that in males (data not shown). Importantly, the amount of RTEC consumed is lower in females than in males as well, reaching significance in teenagers and elderly. Therefore, an increase in the amount of RTEC consumption as a method to improve vitamin D status in females might enhance the benefit even better than in males, especially given the fact that after the NDNS data collection (years 2008–2012) more RTEC variants have been fortified with vitamin D.

The findings of vitamin D intake per day and status (25(OH) D) per age cohort are in good agreement with those by Spiro and Buttriss [33], using the same NDNS database (2008–2012). In both studies a somewhat higher intake of vitamin D was noted in males than in females, except for the elderly. In contrast to our study where the information on status of males and females were combined for power reasons, Spiro and Buttriss distinguished between the genders. In both studies, within the whole population sample, the highest status was observed in children between 4–10 years of age, while their daily intake of vitamin D is relatively low. Therefore, other mechanisms, like UV exposure (such as at playing outside), might be of importance here. As a consequence, since the status is already relatively high in children, an extra intake of 1.0 μg per day will lead to an increase of 1.2 nmol/L when other factors, such as UV exposure, are kept constant, while in teenagers this increase amounts to 2.6, in adults 2.0 and in elderly 2.8 nmol/L. In the latter cohorts there is more room (asymptotically) to improve the status than in that of the children. Overall, it can be concluded that this extra intake is able to increase serum concentrations considerably. 

There are various limitations to the present study. When considering the relationship between actual daily vitamin D intake and 25(OH)D concentrations it is noted that there is a time gap between the recording of vitamin D intake and vitamin D status within the NDNS program, this may be a confounding factor in the model, most likely more affected by a temporal change in sunlight exposure than by a change in actual food intake. Despite this, it was found that the increase in serum concentrations, when actual intake data are being used, is much faster in children than in the elderly. This is reflected in the daily amount of vitamin D intake required to reach 50 nmol/L in serum: from 1.1 μg in children, 4.5 μg in teenagers, 6.1 μg in adults to 7.5 μg per day in the elderly. Bearing this observation in mind fortification in children will reach certain thresholds serum concentrations at lower intake levels than that in the elderly. Importantly, it should be emphasized that 2008–2012 calculated plateau concentrations are not static but a reflection of the current NDNS intake and status data. These plateau concentrations, for example, will increase if in a next NDNS survey higher 25(OH)D concentrations are found in relation to higher intake of vitamin D.

Another limitation is that the extra intake of vitamin D coming from consumption of RTEC, as simulated in the present study, affects the increase in serum 25(OH)D concentrations by the actual amount of RTEC consumed. When only RTEC consumers are analysed and using actual daily RTEC intake data, the impact is substantial as well, resulting in higher vitamin D serum concentrations of ~1.2 nmol/L in children aged 4–10 to ~6.3 nmol/L in the elderly.

The fortification model of daily vitamin D intake and status can be applied to calculate the gain in the latter after fortification with 4.2 μg/100 g RTEC as presented in this manuscript. Unfortunately, the number of observations is low, leading to a relatively large variation. Moreover, the model is based on vitamin D status for all, including people not consuming RTEC. Bearing these conditions in mind and focusing on the RTEC consumers only, the percentage of people predicted to be below 50 nmol/L in the age range between 4 to 10 decreases from about 17% in non-fortified consumers to about 1.5% after fortification with 4.2 μg/100 g RTEC. As expected the percentage of people with vitamin D serum concentrations below 50 nmol/L in the other cohorts will decrease as well: in the 11–18 years old from 92% to 74%, in the 19–64 years old from 83% to 69%, and in the 65 plus years old group from 78% to 58%. These numbers are relatively high and the accuracy in the prediction may improve when the model is based on a higher number of people as presently used. Unfortunately, the low number of observations does not allow appropriate modeling for only the RTEC consumers. 

A third limitation is the impact by ethnicity. In general, darker-skinned persons show lower concentrations of 25(OH)D than white-skinned persons [34]. Concentrations are normally low even in darker-skinned individuals who are fully exposed to solar UV at tropical latitudes. Unfortunately, the dataset did not contain information on ethnicity. Fortification could lead to unhealthy 25(OH)D concentrations depending on the pigmentation of the skin without realizing. Ranges of 25(OH)D being low in white-skinned persons might be optimal in darker-skinned persons [35]. It is emphasized that when available ethnicity should be evaluated as confounding factor in the model as well.

Finally, considerations need to be given to the fact that this analysis was performed on the data from the first four years (2008–2012) of the NDNS survey, and that data for years (5–8) of the NDNS program has become available. Therefore, studies comparing our findings to those and modeling with more recent survey data would prove interesting as they may be important to maintain safe vitamin D fortification practices or may highlight that further fortification vehicles should be identified, especially since the intake of vitamin D, partly due to fortification, might prove to be increased as compared to the levels as used in the present manuscript.

Ready-to-eat cereals are considered a practical vehicle and are already used for vitamin D fortification. Other food items that are frequently consumed by a large proportion of the population could also be applied for this purpose [36]. Ready-to-eat-breakfast cereal, being eaten by a large proportion of the population, offers a suitable vehicle for vitamin D fortification when limiting factors such as ethnicity, toxicity, and stability are met. 

## Figures and Tables

**Figure 1 nutrients-12-01868-f001:**
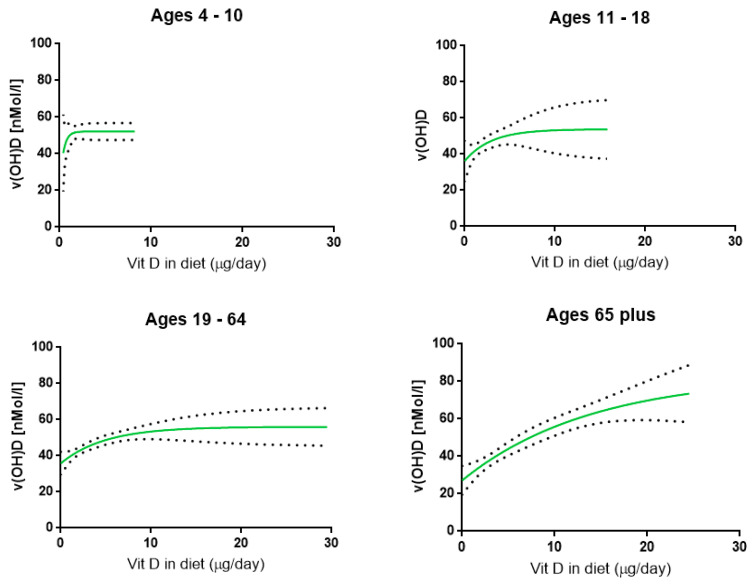
The association between the vitamin D in diet (μg per day) and the serum concentration of 25(OH)D (nmol/L) as observed in the NDNS database. The solid line represents the 50% relationship while the dotted lines show the 95% CI level.

**Figure 2 nutrients-12-01868-f002:**
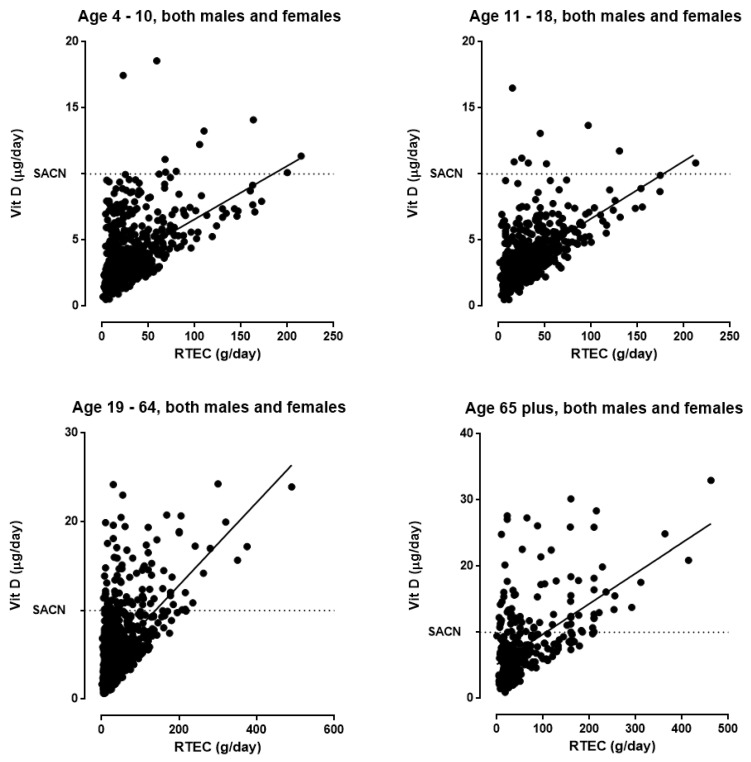
The association between RTEC intake (g/day) and expected vitamin D intake (μg per day) after fortification of 4.2 μg of vitamin D per 100 g of RTEC applying the presented model. The solid line represents the regression line, while the dotted line represents the recommended UK SACN levels of vitamin D intake. Each symbol represents a single person.

**Table 1 nutrients-12-01868-t001:** Intake of vitamin D (μg/day) with or without supplementation per age cohort per gender (data from the NDNS database before application of the model). Total: all persons with and without supplementation.

Age Cohort	Supplement	Males				Females			
		Mean	s.e.	s.d.	*n*	Mean	s.e.	s.d.	*n*
4–10	No	2.03	0.06	0.33	348	1.89	0.06	1.08	336
	Yes	5.92	1.05	2.70	66	7.27	n.a.	10.63	53
	Total	2.65	0.10	2.03	414	2.62	0.22	4.42	389
11–18	No	2.35	0.06	1.30	418	1.90	0.06	1.21	416
	Yes	6.11	n.a.	3.31	27	6.04	n.a.	2.54	23
	Total	2.58	0.08	1.74	445	2.12	0.07	1.60	439
19–64	No	3.13	0.09	2.18	613	2.50	0.07	1.80	769
	Yes	9.43	n.a.	8.35	97	7.80	0.29	3.98	176
	Total	3.99	0.16	4.27	710	3.49	0.11	3.14	945
65 plus	No	3.85	0.21	2.57	146	2.76	0.12	1.57	160
	Yes	8.84	n.a.	5.34	45	9.73	0.64	5.42	77
	Total	5.02	0.29	4.02	191	5.02	0.32	4.67	237

s.e.: standard error. s.d.: standard deviation. n.a.: not applicable due to strata with single sample unit.

**Table 2 nutrients-12-01868-t002:** Vitamin status (25 hydroxy vitamin D serum concentrations (nmol/L)) in the various age cohorts with males and females combined, data from the NDNS database before application of the model.

Vitamin D Status in nmol/L								
					% Below		% Above	
Age Cohort	Mean	s.e.	s.d.	*n*	25 nmol/L	50 nmol/L	75 nmol/L	100 nmol/L
4–10	51.46	1.72	21.83	167	10	48	9	4
11–18	44.90	1.45	24.41	311	21	65	12	3
19–64	45.24	1.00	24.63	788	23	61	12	3
65 plus	43.47	1.68	22.26	201	23	65	10	2

s.e.: standard error. s.d.: standard deviation. *n*: number of persons.

**Table 3 nutrients-12-01868-t003:** Relationship between the amount of vitamin D consumed per day (*x*: in μg per day) and serum concentrations of 25(OH)D (*y*; in nmol/L). Data from the NDNS database before application of the model.

Age Cohort	Formula	Plateau	95 % CI Plateau	*n*
4–10	*y* = 21.38 + 30.77(1−e^−2.42*x*^)	52.15	47.54–56.76	165
11–18	*y* = 36.15 + 17.61(1−e^−0.34*x*^)	53.78	36.55–71.01	311
19–64	*y* = 35.78 + 20.26(1−e^−0.20*x*^)	56.04	45.30–66.79	787
65 plus	*y* = 27.10 + 55.92(1−e^−0.07*x*^)	83.02	39.35–126.70	201

**Table 4 nutrients-12-01868-t004:** Relationship between month of the year (*x*) and serum concentrations of 25(OH)D (*y* in nmol/L), stating the month as an integer. January: 1, February: 2, etc. The time of the year at which the highest concentration of 25(OH)D in serum is found, is expressed as month in the year. Mathematical modeling on the data from the NDNS database.

Cohort	Gender	*n*	Formula	Peak Concentration Reached at
4–10	males	93	*y* = −0.76*x*^2^ + 10.97*x* + 22.71	7.2
4–10	females	74	*y* = −0.43*x*^2^ + 7.80*x* + 24.08	9.1
11–18	males	169	*y* = −0.86*x*^2^ + 13.13*x* + 7.49	7.6
11–18	females	142	*y* = −0.46*x*^2^ + 6.94*x* + 23.09	7.5
19–64	males	333	*y* = −0.65*x*^2^ + 10.58*x* + 10.38	8.1
19–64	females	455	*y* = −0.53*x*^2^ + 8.35*x* + 21.16	7.9
65–90	males	90	*y* = −0.34*x*^2^ + 5.25*x* + 30.68	7.7
65–94	females	111	*y* = −0.47*x*^2^ + 6.87*x* + 23.14	7.3
Overall	males + females	1467	*y* = −0.52*x*^2^ + 8.36*x* + 20.65	8.1

**Table 5 nutrients-12-01868-t005:** Calculated (via mathematical modeling) serum concentrations of 25(OH)D after consumption of fortified ready-to-eat cereals (RTEC) by 4.2 μg/100 g, applying the amount consumed as documented in the NDNS database, keeping the amount of UV exposure constant. Only RTEC consumers used.

Cohort	Gender	With Fortification			Without Fortification				Difference	
		nmol/L			nmol/L			*n*	nmol/L	nmol/L
		Mean	s.e.	s.d.	Mean	s.e.	s.d.		Mean	s.d.
4–10	males	52.02	0.02	0.33	50.98	0.13	2.41	370	1.04	2.30
	females	51.91	0.05	0.85	50.66	0.17	3.06	338	1.26	2.69
11–18	males	48.61	0.17	2.66	45.40	0.16	3.14	307	3.21	2.13
	females	47.04	0.18	3.04	44.36	0.20	3.46	291	2.68	1.98
19–64	males	48.09	0.18	3.74	45.48	0.20	4.22	426	2.61	2.21
	females	47.56	0.18	4.16	44.85	0.20	4.60	606	2.70	2.64
65 plus	males	50.84	n.a.	10.07	43.86	n.a.	9.16	130	6.98	6.51
	females	47.93	0.80	10.28	42.38	0.82	10.51	189	5.55	5.32

*n*: number of persons. s.e.: standard error. s.d.: standard deviation. n.a.: not applicable due to strata with single sampling unit.

**Table 6 nutrients-12-01868-t006:** The percentage of persons with and without fortification with 4.2 μg vitamin D per 100 g RTEC above certain threshold levels of 25(OH)D. Results of mathematical modeling. Please note that only RTEC users have been applied. *n*: total number of persons.

			Percentage of Persons Consuming RTEC			
Age Cohort	Gender	*n*	With Fortification		Without Fortification	
			>75 nmol/L	>100 nmol/L	>75 nmol/L	>100 nmol/L
4–10	males	370	9.3	2.3	9.1	2.3
	females	338	6.4	6.4	6.3	6.3
11–18	males	307	15.0	2.7	14.1	2.5
	females	291	9.6	1.2	9.1	1.1
19–64	males	426	11.7	2.0	11.1	1.9
	females	606	11.8	5.0	11.1	4.8
65 plus	males	130	18.1	1.8	15.6	1.6
	females	189	11.6	1.4	10.3	1.3

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
