# Peer review of "Vitamin D Serum Levels in the UK Population, including a Mathematical Approach to Evaluate the Impact of Vitamin D Fortified Ready-to-Eat Breakfast Cereals: Application of the NDNS Database"

_nutrients, 2020, doi:10.3390/nu12061868_

Round 1

Reviewer 1 Report

This manuscript reports on the relationship between vitamin D status and intake in the  UK population and the impact of vitamin D fortified ready-to-eat cereals on this status via  data acquired from the National Diet and Nutrition Survey. This is an excellent manuscript on a very important topic of great interest for the UK Public Health.  It should attract the interest of a wide audience. It is well written and professionally presented. I recommend publication as it states.

Author Response

First of all, the authors would like to thank both the editor and the reviewers for their time, but also for their constructive remarks. We do believe that after modification of our manuscript the content is more balanced than before and put more into perspective with respect to the target population: the overall UK participant.

The authors have modified a substantial part of The Introduction and especially The Discussion to accommodate the various remarks by the reviewers. During this work some small errors were encountered and corrected as well.

Reviewer 1.

Obviously the authors would like to thank the reviewer for her/his very positive remarks. We agree that the content of our manuscript is of importance to the UK population, especially regarding minimizing health risks. Hopefully our contribution will be a small step to optimize the health status in the above mentioned population.

The authors hope to have reacted on the various remarks by the various reviewers to their satisfaction. However we understand that in one manuscript one cannot offer a full picture on this important topic, especially not when factors, such as ethnicity and type of vitamin D provided, are lacking in the database.

The authors are looking forward to the reaction of the editor and the reviewers. In the meantime we are open to any issue there might be.

Kind regards, also on behalf of Laura Street and Toine Hulshof,

Wim Calame,

Corresponding author.

Reviewer 2 Report

The study focuses on a relevant challenge for human nutrition that is the deficiency in the intake of vitamin D. The Authors applied an original modelling approach and achieved new information.

I found some major flaws in the study.

First of all, the source of vitamin D in the fortified products was not reported (lines 127-129), while this information is necessary.

Second, the Authors explained the reasons why cereal breakfast were chosen as a vehicle food for fortification. Among these reasons, considerations about vitamin D stability in food are completely lacking (lines 66-78). Vitamin D stability is largely affected by the water activity, with the degradation rate increasing with increasing the water activity. In fact, storage at aw 0.11 resulted in a half-life of 225 days at 30 °C, while storage at aw 0.75 at 30 °C resulted in half-life of 57 days (see: Pedrali, D., Gallotti, F., Proserpio, C., Pagliarini, E., Lavelli, V. Kinetic study of vitamin D2 degradation in mushroom powder to improve its applications in fortified foods. LWT - Food Sci. Technol., 2020, 125, 109248)

Third, the number of persons involved in the survey should be stated in the abstract and in the tables.

Author Response

First of all, the authors would like to thank both the editor and the reviewers for their time, but also for their constructive remarks. We do believe that after modification of our manuscript the content is more balanced than before and put more into perspective with respect to the target population: the overall UK participant.

The authors have modified a substantial part of The Introduction and especially The Discussion to accommodate the various remarks by the reviewers. During this work some small errors were encountered and corrected as well.

Reviewer 2.

The reviewers would like to thank this reviewer, especially with respect to her/his statement on stability of the vitamin D in the fortified products. Clearly the authors have underestimated the importance of this topic in the content of our message.

  • Source of the fortified products as documented in the NDNS database.

The reviewer is right on this. In the revised manuscript we have stated that the vitamin D fortification in the NDNS data base was observed in biscuits, cereal products (including chocolate-enriched products) and dairy products (ranging from low-fat butter to yoghurt) at lines 86-91, without specification on the type of vitamin D. The data base was once again checked for additional information on the type of vitamin D enrichment (ergocalciferol (D2) versus cholecalciferol (D3)) but again observed that this was unfortunately not present. Since we have used real data in all our calculations we have used the same (unknown) relative abundance of both types as present in the database. Due to your relevant remark we have mentioned this fact in the Introduction (lines 86-91) and have put this into perspective in the Discussion (lines 364-370). I (Wim Calame) have taken the liberty to use the reference as mentioned in your text.

  • Stability of vitamin D in fortified products.

Indeed, there is no mentioning of vitamin D stability in our fortified ready-to-eat breakfast cereals (RTEC). The reason is that the authors were convinced that since the products concern a dry application, this wouldn’t be an issue. The reviewer is right that we should have discussed the stability anyhow. Please find the topic in the Introduction in lines 76-82 of the revised manuscript.

  • Numbers of persons in abstract and tables.

Due to minimizing the word count in the abstract there was no mentioning of the numbers of persons from the NDNS database in this section. But then, to be honest, we did not mention this in the main body of text either. We have corrected this in lines 17 and 18 of the Abstract (after taking away another line to avoid too many words in the Abstract). We have inserted the same information in lines 98 and 99 of the Introduction and in lines 121-122 of the Materials and Methods. With respect to the various Tables, only Table 4 lacked the various numbers of included persons. We have corrected this on page 8.

The authors hope to have reacted on the various remarks by the various reviewers to their satisfaction. However we understand that in one manuscript one cannot offer a full picture on this important topic, especially not when factors, such as ethnicity and type of vitamin D provided, are lacking in the database.

The authors are looking forward to the reaction of the editor and the reviewers. In the meantime we are open to any issue there might be.

Kind regards, also on behalf of Laura Street and Toine Hulshof,

Wim Calame,

Corresponding author.

Reviewer 3 Report

Variability of vitamin D serum levels

The UK population shows a high variability of vitamin D serum levels. For the 19-64 age cohort (Table 2), the standard deviation is 24.63, with almost a quarter of the population below 25 nmol/L.  If the distribution is normal, another quarter would be above 75 nmol/L, but the distribution probably is not normal. It is almost certainly skewed to the left, and the proportion above 75 nmol/L should be larger than a quarter.

Ethnicity

How can we explain this variability? One factor is age, as the authors note. Another is ethnicity, specifically whether one is of Asian or African/Afro-Caribbean descent. Both groups have lower serum levels of vitamin D. This point is reviewed in the introduction to Farrar et al. (2014):

There is growing evidence of vitamin D insufficiency and deficiency in the United Kingdom (12-14). Low vitamin D status is particularly prevalent in darker-skinned people, with many reports of low concentrations and related health problems in South Asians (15-20). Like many countries at a similar latitude (50-60°N), the United Kingdom has a significant and rising population of individuals of sun-reactive skin type V (ie, with brown skin) who are particularly of South Asian ethnicity (21, 22). Although South Asians reportedly have the same capacity to synthesize vitamin D as do whites (sun-reactive skin types I-IV), pigmented skin requires greater sunlight exposure to raise circulating 25(OH)D as melanin absorbs a proportion of the incident UVB (23-25).

Vitamin D insufficiency and deficiency in the South Asian community used to be attributed to diet or sun avoidance. In reality, as Farrar et al. (2014) show, regular sun exposure at UK latitudes cannot produce sufficient vitamin D in the skin of individuals of South Asian ethnicity.

At the time of the 2011 Census, over 10% of the UK population was of non-European descent. That proportion is approximately 15% today. Ethnicity should therefore explain much of the variability in vitamin D serum levels. In particular, the proportion of individuals below 25 nmol/L should be larger among darker-skinned citizens of the UK. How much larger? Unfortunately, the National Diet and Nutrition Survey does not break down its data by ethnicity. This is a shortcoming in the data that undermines the authors’ conclusions and recommendations.

In my opinion, vitamin D supplementation is not justified, certainly not for the entire UK population. One could argue for an approach that is targeted at those minority groups that are known to be vitamin-D insufficient or deficient. There are problems with that approach, however, as I will explain.

Risk of vitamin D toxicity

Vitamin D is fat-soluble and accumulates in the body. This is not the case with vitamin C, which is water-soluble and is excreted if any excess develops. Nonetheless, an excess of vitamin D does not develop in the body under natural conditions. Prolonged exposure to the sun cannot lead to an excess, since prolonged sun exposure also degrades previtamin D3, principally to lumisterol. This control mechanism does not exist with vitamin D supplementation.

The authors acknowledge that supplementation can lead to toxic levels of vitamin D, but they reassure the reader that this is not "an issue". They are right for the most obvious effects of vitamin D toxicity, notably hypercalcemia. Other effects, however, can arise.

We know that a U-shaped response curve describes the relationship between serum 25(OH)D and various disease risks (Schroff et al. 2010). According to a Finnish study, the risk of prostate cancer increases below 40 nmol/L and above 60 nmol/L (Tuohimaa 2009; Tuohimaa et al. 2009). A Swedish cohort study found that a 50% higher mortality rate is associated with concentrations of serum 25(OH)D below 46 nmol/L and above 98 nmol/L (Michaëlsson et al. 2010). In women from the United States, Finland, and China, mortality for seven types of cancer (endometrial, esophageal, gastric, kidney, non-Hodgkin's lymphoma, pancreatic, ovarian) increases below 45 nmol/L and above 124 nmol/L (Helzlsouer & Steering Committee 2010). Another transnational study reported a higher risk of pancreatic cancer above 100 nmol/L (Stolzenberg-Solomon et al. 2010). The Framingham Heart Study concluded that cardiovascular disease risk increases below 50 nmol/L and above 62.5 nmol/L, while the NHANES III found higher all-cause mortality above 122.5 nmol/L (Davis 2009). Perhaps most worrisome, animal and human studies have revealed a U-shaped response curve for lifespan, with premature aging associated with either too little or too much vitamin D (Tuohimaa 2009; Tuohimaa et al. 2009).

Optimal vitamin D levels thus span a narrower range of values than the authors may think. A higher risk of prostate cancer becomes measurable even at 60 nmol/L, and a rise in overall mortality becomes measurable at 100 nmol/L. For this reason, the authors should estimate the impact of their dietary supplementation proposal in terms of the increased proportion of the population with high vitamin D levels, specifically the proportions above 75 and 100 nmol/L.

Ethnic differences in optimal vitamin D levels

Vitamin D metabolism has coevolved with ambient levels of UV light and with darkness of skin pigmentation. This coevolution has been studied among the Inuit people of Alaska, northern Canada and Greenland (Frost 2012; Frost 2018). Above the Arctic Circle, solar UV is usually too weak to permit vitamin D synthesis in the skin. Some vitamin D is obtained through consumption of fatty fish and certain marine mammals, but those sources are insufficient. According to a dietary study conducted in Nunavut and the Northwest Territories, the median vitamin D intake was 5.13±5.34 µg/day for Inuit on a traditional diet (=300 g of fish or game meat per day) and 3.5±3.22 µg/day for Inuit on a non-traditional diet (<300 g). Both groups had much less than the Reference Nutrient Intake of 10 µg/day (Kolahdooz et al. 2013).

Inuit have coped with this chronic vitamin D insufficiency through physiological changes: receptors that bind more tightly to the vitamin D molecule; a lower set-point for calcium-regulated release of parathyroid hormone; and conversion of vitamin D at a higher rate from its common form to its most active form. There may be other physiological adaptations. For instance, Inuit breast milk might be richer in ß-casein, which seems to facilitate the body's use of vitamin D (Frost 2018).

There has also been coevolution between vitamin D metabolism and dark skin (Frost 2012). Vitamin D levels are generally lower in dark-skinned populations, even in tropical regions with high solar UV. A study from south India found levels below 50 nmol/L in 44% of the men and 70% of the women. The subjects were "agricultural workers starting their day at 0800 and working outdoors until 1700 with their face, chest, back, legs, arms, and forearms exposed to sunlight" (Harinarayan et al. 2007). In a study from Saudi Arabia, levels were below 25 nmol/L in respectively 35%, 45%, 53%, and 50% of normal male university students of Saudi, Jordanian, Egyptian, and other origins (Sedrani 1984). In a sample of healthy Middle Eastern athletes, 91% had levels below 50 nmol/L (Hamilton et al. 2010). There is significant variation even among Europeans, with levels being lower in central and southern Europeans than in lighter-skinned Swedes (Snellman et al. 2009). Finally, a meta-analysis concluded that serum 25(OH)D levels are significantly lower in people of non-European origin than in people of European origin. In the first group, the levels are consistently low regardless of latitude (Hagenau et al. 2009).

In darker-skinned populations, the optimal range of vitamin D serum levels seems to be shifted toward lower values. A sample of African Americans showed a positive correlation between calcified plaque formation (atherosclerosis) and serum 25(OH)D, despite a negative correlation among European Americans over the same range (Freedman et al. 2010). Similarly, vitamin-D supplementation did not help reduce bone loss or increase bone turnover in postmenopausal African-American women, even though their mean serum 25(OH)D was only 47 nmol/L before supplementation (Aloia et al. 2005).

We define "healthy vitamin D status" on the basis of data from light-skinned populations of European descent. We may thus categorise healthy dark-skinned people as vitamin-D deficient, simply because their levels of serum vitamin D are low by our standards.  For such individuals, supplementation may have the unintended effect of pushing them into the zone of vitamin D toxicity.

Recommendations for the manuscript.

  1. The authors should modify Table 1 to provide information on the current proportions of individuals with serum levels above 75 nmol/L and 100 nmol/L.

  1. A second table, modelled on Table 1, should be provided to show how the proposed dietary supplementation would change the distribution of vitamin D serum levels. Specifically, what proportion of the UK population would be above the thresholds of 75 nmol/L and 100 nmol/L?

  1. The authors should note that serum levels are normally lower in dark-skinned individuals and, conversely, higher in light-skinned individuals. They should further acknowledge that lower vitamin D levels in darker-skinned groups may be physiological normal. At the very least, research is needed to calculate optimal vitamin D levels for such groups on the basis of data from the same groups.

References

Aloia, J.F., S. A. Talwar, S. Pollack, & J. Yeh. (2005). A randomized controlled trial of vitamin D3 supplementation in African American women. Arch Intern Med 165:1618-1623.

Davis CD. (2009). Vitamin D and health. Can too much be harmful? Am J Lifestyle Med 3(5):407-408.

Farrar, M. D., R. Kift, S. J. Felton, J. L. Berry, M. T. Durkin, D. Allan, A. Vail, A. R. Webb, and L. E. Rhodes. (2014). Recommended summer sunlight exposure amounts fail to produce sufficient vitamin D status in UK adults of South Asian origin. The American Journal of Clinical Nutrition 94(5): 1219-1224.

Freedman, B.I., L. E. Wagenknecht, K. G. Hairston et al. (2010). Vitamin D, adiposity, and calcified atherosclerotic plaque in African-Americans. J Clin Endocrinol Metab 95:1076-1083.

Frost, P. (2012). Vitamin D deficiency among northern Native Peoples: a real or apparent problem? International Journal of Circumpolar Health 71: 18001

Frost, P. (2018). To supplement or not to supplement: are Inuit getting enough vitamin D? Études Inuit Studies 40(2): 271-291.

Hagenau, T., R. Vest, T. N. Gissel, et al. (2009). Global vitamin D levels in relation to age, gender, skin pigmentation and latitude: an ecologic meta-regression analysis. Osteoporos Int 20:133-140.

Hamilton, B., J. Grantham, S. Racinais, & H. Chalabi. (2020). Vitamin D deficiency is endemic in Middle Eastern sportsmen. Public Health Nutr 13:1528-1534.

Harinarayan, C.V., T. Ramalakshmi, U. V. Prasad, et al. (2007). High prevalence of low dietary calcium, high phytate consumption, and vitamin D deficiency in healthy south Indians. Am J Clin Nutr 85:1062-1067.

Helzlsouer K. J. & Steering Committee Vitamin D Pooling Project of Rarer Cancers. (2010). Vitamin D: Panacea or a Pandora's box for prevention? Cancer Prev Res 3(1 Suppl):PL04-05.

Kolahdooz, F., A. Barr, C. Roache, T. Sheehy, A. Corriveau, & S. Sharma. (2013). Dietary adequacy of vitamin D and calcium among Inuit and Inuvialuit women of child-bearing age in Arctic Canada: A growing concern. PLoS One 8(11):

Michaëlsson, K., J. A. Baron, G. Snellman, R. Gedeborg, L. Byberg, J. Sundström, L. Berglund, J. Ärnlöv, P. Hellman, R. Blomhoff, A. Wolk, H. Garmo, L. Holmberg, & H. Melhus. (2010). Plasma vitamin D and mortality in older men: a community-based prospective cohort study. The American Journal of Clinical Nutrition 92(4): 841-848,

Sedrani, S.H. (1984). Low 25-hydroxyvitamin D and normal serum calcium concentrations in Saudi Arabia: Riyadh region. Ann Nutr Metab 28:181-185.

Shroff, R., C. Knott, & L. Rees. (2010). The virtues of vitamin D-but how much is too much? Pediatr Nephrol 25:1607-1620.

Snellman, G., H. Melhus, R. Gedeborg, et al. (2009). Seasonal genetic influence on serum 25-hydroxyvitamin D levels: a twin study. PLoS ONE 4(11): e7747.

Stolzenberg-Solomon, R. Z., E. J. Jacobs, A. A. Arslan, et al. (2010). Circulating 25-hydroxyvitamin D and risk of pancreatic cancer. Cohort Consortium Vitamin D Pooling Project of Rarer Cancers. Am J Epidemiol 172:81-93.

Tuohimaa, P. (2009). Vitamin D and aging. J Steroid Biochem Mol Biol 114:78-84.

Tuohimaa, P., T. Keisala, A. Minasyan, J. Cachat, & A. Kalueff. (2009). Vitamin D, nervous system and aging. Psychoneuroendocrinology 34S:S278-S286

Author Response

First of all, the authors would like to thank both the editor and the reviewers for their time, but also for their constructive remarks. We do believe that after modification of our manuscript the content is more balanced than before and put more into perspective with respect to the target population: the overall UK participant.

The authors have modified a substantial part of The Introduction and especially The Discussion to accommodate the various remarks by the reviewers. During this work some small errors were encountered and corrected as well.

Reviewer 3.

The authors are very grateful for the very thorough way this reviewer has brought various points of interest to our attention. This is particularly true for the potential toxicity of a too high concentration of serum levels of vitamin D as well as the impact by ethnicity. The former was already addressed in the Discussion but more emphasis should be laid on the fact that too low concentrations can be detrimental as well, please see lines 386-389. Since the latter (ethnicity) was not recorded in the data set we were unfortunately unable to analyze this parameter on the outcome of our study in contrast to age and gender despite the fact that we were aware of this issue. Therefore we omitted this important topic for discussion in the present manuscript. The reviewer is absolutely right on the impact of the above. Moreover we thank her/him for the thorough explanation she/he has given in his reviewing process! Very noteworthy. We have used various references as suggested.

  • Variability of vitamin D serum levels.

To address this issue we have made an extension of Table 2 by providing the percentages of persons above 75 nMol/L and that of persons above 100 nmol/L . Please check Table 2 on page 6. Please acknowledge the fact that the number of persons is relatively low and that the population consists of both ready-to-eat cereals (RTEC) consumers and non-consumers. This is of importance with respect to the newly made Table 6 (page 11) on the percentage of persons above 75 nmol/L and above 100nmol/L to reflect the presented model. Since we are discussing the serum levels more thoroughly in the Discussion section the new Table 6 (on levels above 75 and 100 nmol/L) is put in that section to avoid disruption of the logics of our message (lines 386-399).

  •  

The reviewer is right. Ethnicity is believed to play a major role in the serum levels of vitamin D. Unfortunately we couldn’t obtain any information to evaluate this relationship. The authors were aware of the impact of ethnicity on vitamin D serum levels, but since this issue couldn’t be addressed we did not discuss it in the discussion. Unfortunately we should have done so. Therefore we have discussed this point at various positions in the Discussion (lines 466-470, 486-488).

  • Conclusions as stated in the manuscript.

The reviewer is right that we should introduce some more details in the approach to optimize vitamin D levels in the UK population. Hopefully the changes in the text , especially in the Discussion section, give credit to the interfering confounding factors. The authors have tried to minimize the addition of information in the Discussion because this section was already relatively long. Hopefully the reviewers will appreciate this, as well as the removal of some “old” sentences to avoid a too lengthy discussion.

The authors hope to have reacted on the various remarks by the various reviewers to their satisfaction. However we understand that in one manuscript one cannot offer a full picture on this important topic, especially not when factors, such as ethnicity and type of vitamin D provided, are lacking in the database.

The authors are looking forward to the reaction of the editor and the reviewers. In the meantime we are open to any issue there might be.

Kind regards, also on behalf of Laura Street and Toine Hulshof,

Wim Calame,

Corresponding author.

Round 2

Reviewer 2 Report

The Authors fully addressed the points raised by this reviewer

Author Response

The authors appreciate the effort made by the reviewer in optimizing the content of our manuscript. We are pleased to understand that this was done according to her/his liking.

Reviewer 3 Report

I would recommend the following changes and insertions:

Lines 386-388 - replace the sentence with the following: "High concentrations of serum 25(OH)D are as problematic as low concentrations. A Swedish cohort study found a 50% higher mortality rate associated  with concentrations below 46 nmol/L and above 98 nmol/L. This U-shaped response curve has been noted by other studies." 

Add reference to Michaelsson et al. (2010) here and at the end.

Line 467 - insert the following sentence: "Concentrations are normally low even in darker-skinned individuals who are fully exposed to solar UV at tropical latitudes"

Line 599 - insert Michaelsson et al. (2010)

Michaëlsson, K., J. A. Baron, G. Snellman, R. Gedeborg, L. Byberg, J. Sundström, L. Berglund, J. Ärnlöv, P. Hellman, R. Blomfoff, A. Wolk, H. Garmo, L. Holmberg, H. Melhus. Plasma vitamin D and mortality in older men: a community-based prospective cohort study. Am J Clin Nutr 2010, 92, 841-848.  

Author Response

Upon the advice given by the reviewer the authors have made the changes as suggested and have inserted the reference accordingly. The main issue is the awareness that the lower and upper threshold levels of healthy concentrations of 25(OH)D are different between darker- and white-skinned consumers and that this phenomenon should be considered when fortification is performed. Indeed this is an important issue within the vitamin D discussion, as put forward by the reviewer and we thank her/him for this.